# Automated Think-Aloud Protocol for Identifying Students with Reading Comprehension Impairment Using Sentence Embedding

**Yongseok Yoo** 

School of Computer Science and Engineering, Soongsil University, Seoul 06978, Republic of Korea;
yyoo@ssu.ac.kr; Tel.: +82-2-820-0678

**Abstract:** The think-aloud protocol is a valuable tool for investigating readers' cognitive processes during reading. However, its reliance on experienced human evaluators poses challenges in terms of efficiency and scalability. To address this limitation, this study proposes a novel application of natural language processing to automate the think-aloud protocol. Specifically, we use a sentence embedding technique to encode the stimulus text and corresponding readers' responses into high-dimensional vectors, and the similarity between these embeddings serves as a feature. The properties of the feature are investigated for word-frequency-based and contextualized embedding models. Differences in the sentence embedding-based feature between poor comprehenders and normal readers are investigated. Using these features, seven machine learning models were trained to classify readers into normal and abnormal groups. The highest F1 score of 0.74 was achieved with the contextualized embedding and random forest classifier. This highlights the effectiveness of the embedding technique in extracting useful features for automating the think-aloud protocol for assessing reading comprehension abilities. The potential benefits of this automated approach include increased efficiency and scalability, ultimately facilitating the early diagnosis of reading comprehension impairment and individualized interventions.

**Keywords:** reading comprehension; think-aloud; natural language processing; sentence embedding



## 1. Introduction

### 1.1. Reading Comprehension Impairment

There have been persistent efforts to facilitate the diagnosis of a reading disorder in order to provide early interventions for students at a young age. The task of diagnosing a reading disorder is challenging because reading involves the complex interplay of multiple cognitive processes, such as letter–sound correspondence, phonological memory, word recognition, sentence processing, and comprehension [1]. A deficiency in any of these fundamental processes could give rise to reading disorders [2,3]. For instance, deficits in lexicography and phonics could lead to difficulties in word identification and dyslexia [2,3]. In contrast, higher-level cognitive processes such as reading comprehension are less understood [3].

Reading comprehension impairment, also known as specific reading comprehension disorder, is a specific learning disorder characterized by persistent difficulties in understanding meaning from written text [2,3]. It should be distinguished from dyslexia, where the main difficulty lies in accurate word decoding. In the case of reading comprehension impairment, individuals typically possess adequate word decoding skills but encounter difficulties in understanding text.

Early diagnosis is crucial for enabling timely interventions and providing essential support to address the specific needs of individuals with reading comprehension impairment [4]. Typically, the identification of poor comprehenders involves a two-step process.

Initially, teachers observe students with low classroom performance and try to pinpoint the specific reasons behind their difficulties. Subsequently, students at risk are referred to experts, such as educational psychologists and reading specialists, who conduct formal assessments and diagnose the underlying difficulties. This collaborative approach necessitates diligent monitoring, expertise in reading comprehension, and the allocation of appropriate resources.

### 1.2. Think-Aloud Protocol

The think-aloud protocol is a powerful qualitative technique for the assessment of the cognitive processes involved in reading comprehension [5,6]. The think-aloud protocol is administered as follows. While performing a given task, the subjects verbalize their thoughts. The utterances are transcribed and analyzed by the experts. The think-aloud protocol allows the researchers to gain insight into the cognitive processes of the subjects and to identify any reading difficulties that may exist. Thus, the think-aloud protocol is an effective tool for diagnosing reading difficulties.

The advantages of the think-aloud protocol are as follows. First, the think-aloud protocol provides direct access to a person's thoughts and reasoning while engaged in a task. Second, think-aloud provides moment-to-moment monitoring of cognitive processes that may be missed in other retrospective assessments. Third, the think-aloud protocol provides rich qualitative data that can be analyzed to uncover patterns and strategies employed by readers. This qualitative assessment complements quantitative measures and can provide a more comprehensive understanding of how subjects read a given text.

However, the richness of the data collected using the think-aloud protocol presents several challenges. First, the diversity of responses and subjective interpretation limits the generalizability and replicability of studies using the think-aloud protocol [7]. Unlike other evaluations such as surveys or questionnaires, user responses in the think-aloud protocol are open-ended. Therefore, categorizing and interpreting user responses becomes more complex and subjective. Researchers must navigate through a variety of perspectives and different ways of expressing thoughts. Therefore, different research groups have different criteria, and the same response may be interpreted differently by evaluators. The categorization and interpretation process may require iteratively refining coding schemes, consulting with multiple experts to establish consensus, and ensuring transparency. Moreover, the sheer volume of data generated by think-aloud sessions can be overwhelming. Transcribing and analyzing large amounts of verbal data require significant time and resources.

### 1.3. Contributions of this Study

This study aims to overcome such limitations of the think-aloud protocol by using natural language processing (NLP). NLP is a field of artificial intelligence that focuses on developing computational models for understanding human language [8]. One of the important NLP techniques used in this study is text embedding, which involves representing a word or sentence as a fixed-length vector [9–16]. This numerical representation is used for downstream tasks. Recently, neural network-based embedding models that take contextual information into account are being trained using a large corpus [11–16], which has greatly advanced NLP applications.

The key contributions of this study are as follows. First, we propose a method to represent the relationships between the stimulus text and think-aloud responses (Figure 1). For this, sentences of stimulus texts and subjects' responses are first encoded to high-dimensional vectors using sentence embedding techniques. Then, similarity scores between input sentence and readers' responses are calculated. Differences in these similarity scores between the two sentence embedding models are investigated.

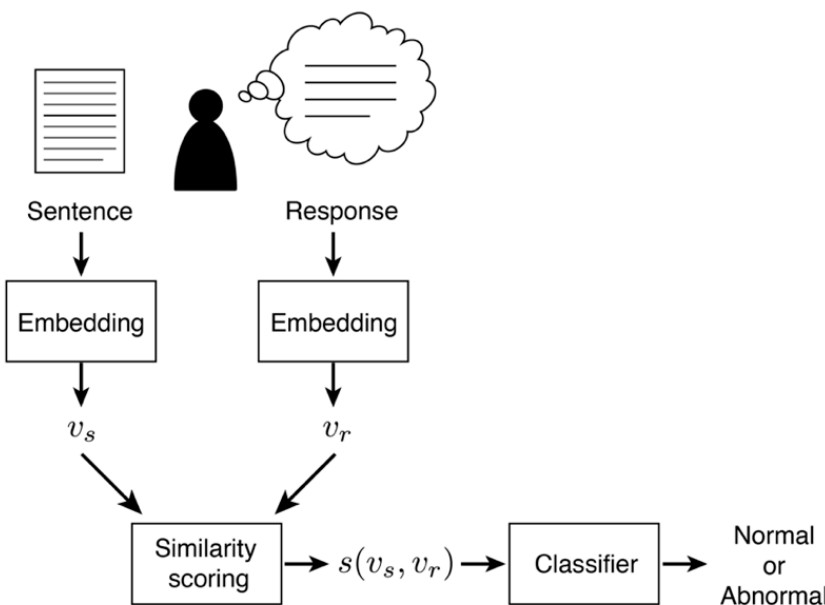

**Figure 1.** Schematic diagram of automated diagnosis of reading comprehension impairment using think-aloud protocol.

Second, we propose to use those embedding-based similarity scores to automatically identify poor comprehenders using the think-aloud protocol. Machine learning models are trained to classify poor comprehenders and normal students from their responses to stimulus texts. The classification accuracy of the proposed method is compared with the diagnoses of human experts.

Therefore, the proposed method aims to address the challenges associated with the think-aloud protocol mentioned above. First, the proposed method improves replicability by providing objective criteria for quantifying readers' responses to text. Second, it could automate the think-aloud protocol and efficiently identify poor comprehenders at a lower cost.

The remainder of this paper is organized as follows. Section 2 describes data collection, sentence embedding-based feature extraction, and classification of normal and abnormal. In Section 3, we examine the differences in the sentence embedding-based features between poor comprehenders and normal readers, and classification accuracies of the classifiers are measured. In Section 4, we discuss the results and their implications. In Section 5, we draw general conclusions with future research directions.

## 2. Materials and Methods

### 2.1. Participants Information

Seventy-two third- and fourth-grade students in 14 public elementary schools in South Korea participated in this study. Among them, 28 poor comprehenders were identified by screening (achievement below the 16th percentile on the district-wide reading assessments), followed by the standardized reading assessment battery tests of Reading Achievement and Reading Cognitive Process (RA-RCP) tests [17]. These students with reading comprehension impairments had intact word-recognition capabilities and communication skills. The remaining 44 average students were used as a control group.

### 2.2. Data Collection

In the training session, three sentences were presented, and the participants were asked to verbalize their thoughts right after reading each sentence.

The stimulus text for the main session was adopted from a textbook for elementary reading courses. The text included 10 sentences, and the average number of words in

each sentence was 9.2 with a standard deviation of 2.7. The participants verbalized their thoughts right after reading each sentence, and those responses were transcribed.

As a result, the dataset of 720 pairs of sentences and responses was collected by transcribing the responses of 72 participants to 10 sentences during the main session. The average number of words in each response was 12.8, with a standard deviation of 9.1. Thus, the responses were generally longer than the sentences in the stimulus text. This difference in word length between text and response may be due to the difference between written and oral language.

### 2.3. Feature Extraction

First, each sentence of the stimulus text and corresponding response was encoded to numerical representations using sentence embedding. For the embedding models, term frequency–inverse document frequency (TF-IDF) [10] and Sentence BERT (SBERT) [15,16] were used. The main motivation for choosing these two embedding models was to compare the word-frequency-based and contextualized embeddings. TF-IDF is calculated from the frequencies of words and how insensitive their arrangement is. In contrast, SBERT is the Transformer-based embedding that considers neighboring words using the attention mechanism [15,16].

For TF-IDF, the vocabulary was built by first normalizing the dataset and extracting nouns, similarly to previous studies [18,19], using the open-source Korean text processor [20]. Nouns that appeared more than once in the dataset were collected to form the vocabulary size of 246.

Using TF-IDF, stimulus sentences and readers' responses were encoded to 246 dimensional vectors, with each dimension representing the frequency of the corresponding noun. Specifically, the TF-IDF value is defined by the product of the term frequency (TF) and inverse document frequency (IDF) as follows:

$$\begin{aligned} \text{TF-IDF}(w, t) &= \text{TF}(w, t)\text{IDF}(w) \\ \text{IDF}(w) &= \log \frac{1+N}{1+\text{DF}(w)} + 1 \end{aligned} \tag{1}$$

where TF(w,t) is the term frequency of the given term w in the sentence t, N is the total number of sentences, and DF(w) is the number of sentences that contain term w. The resulting TF-IDF vectors were normalized using the Euclidean norm.

SBERT is a variation in the Transformer architecture [21] tailored for sentence-level representations. Employing the Siamese network structure [15,16], the SBERT model learns sentence embeddings that have smaller distances for semantically similar sentences. To train an SBERT model, sentence pairs with annotated semantic similarity scores are used. This is in contrast to other Transformer models that learn numerical representations by learning to predict masked words based on the surrounding context.

Sentence embedding by SBERT was performed as follows. Following the preprocessing method for the Korean Language Understanding Evaluation (KLUE) benchmark dataset [22], each sentence and its corresponding response were tokenized into morphemes, and byte pair encoding was applied with the vocabulary size of 32 K. A pre-trained SBERT model with RoBERTa architecture [23] for Korean was obtained from [24], which was trained with the KorNLU dataset [25] provided by KakaoBrain [26]. Using this model, 768 dimensional embedding vectors were calculated for all the sentences in the stimulus text and readers' responses.

Next, similarity scores between the embeddings of the stimulus sentences and the corresponding responses were calculated as follows. For each sentence and corresponding response, embedding vectors are called $v_s$ and $v_r$, respectively. The cosine similarity score between these embeddings, called $s(v_s, v_r)$, is calculated as follows.

$$s(v_s, v_r) = \frac{v_s \cdot v_r}{|v_s|\,|v_r|} \tag{2}$$

where $\cdot$ denotes the inner product of two vectors and $|\;|$ represents the Euclidean norm. Because there were 10 sentences in the stimulus text, 10 similarity scores were computed for each participant.

### 2.4. Classification Models

Seven machine learning models were chosen to classify the similarity scores into normal and abnormal classes.

The first three classifiers were simple models, as follows. The first classifier is logistic regression [27], which is simple and effective for linearly separating the input features into two groups. More specifically, the log odds between the two classes are related to a linear function of the input features, resulting in a linear decision boundary between the two classes. The second classifier is Linear Discriminant Analysis (LDA) [28]. LDA is another linear classifier, but based on different assumptions about the input distributions, the input features are normally distributed with different means for different classes and the same covariance. The third classifier is quadratic discriminant analysis (QDA) [29]. Similar to LDA, QDA assumes that the input features are normally distributed with different means for different classes. In contrast, QDA allows the covariances of the classes to be different, resulting in non-linear decision boundaries.

Four more advanced classifiers were chosen to capture more complex patterns in the feature. The Naïve Bayes classifier [30] was used to investigate the effect of potential correlations between the input features. The Naïve Bayes classifier modeled the input features independently of each other, providing a baseline accuracy for ignoring the interactions between input features. Next, a support vector classifier (SVC) with a linear kernel [31] was used to determine the hyperplane that separated the input features with different classes with the maximum margin. In addition, the random forest classifier [32] was used to capture complex and potentially non-linear patterns in the data using an ensemble of randomly selected decision trees. The final model was the k-nearest neighbor (KNN) classifier [33]. KNN could capture more complex and local structures in the data because the class label of the input sample was determined by the class labels of the KNNs.

The hyperparameters of the SVC, random forest, and KNN classifiers varied for a wide range of values, and the highest accuracy was reported for each classifier. In the SVC, the level of robustness to outliers was controlled by the weight (C) of the penalty for misclassification during training. The value of C varied from $10^{-5}$ to $10^{5}$ by the factor of $10^{-0.25}$. The complexity level of the random forest was controlled by the number of estimators (n). Increasing the number of estimators enabled the classifier to consider more complex data patterns. However, excessive estimators would result in overfitting and poor generalization to new inputs. The value of n increased from 2 to 30 in steps of 2. In the KNN classifier, the number of neighbors (k) controlled the smoothness of the decision boundary. The value of k increased from 1 to 25 in steps of 2.

### 2.5. Classification Accuracy Measured Using Cross-Validation

The F1 scores of the seven classifiers for each embedding model were measured using stratified 5-fold cross-validation [34]. Specifically, the dataset was randomly shuffled and divided into five folds so that each fold had the same proportion of normal and abnormal participants. For a given embedding model, a classifier was trained using all but one fold, and the F1 score of the classifier was measured for the omitted fold. This was repeated for all folds.

The statistical significance of the F1 scores was calculated as follows. The baseline is the classifier that always predicts every participant as the majority class (abnormal). The F1 score of this trivial classifier is 0.56. Therefore, the five F1 scores obtained from cross-validation were compared with the baseline F1 score of 0.56 using a one-tailed *t*-test.

### 3. Results

#### *3.1. Similarity Scores*

Different embedding models produced qualitatively different similarity scores. Figure 2 shows histograms of similarity scores between given sentences and corresponding responses based on TF-IDF (A) and SBERT (B). For TF-IDF, the similarity scores of both normal and abnormal groups were bi-modal, with small values close to 0 or larger values close to 1 more frequent (Figure 2A). The median values of the similarity scores were 0.60 (normal) and 0.63 (abnormal). However, this difference in median between the two groups was not statistically significant ($p = 0.07$, Mann–Whitney U test). In contrast, the similarity scores based on SBERT were clustered around higher values for both normal and abnormal groups (Figure 2B). The abnormal group tended to have lower similarity scores (mean = 0.67) than the normal group (mean = 0.77). This difference was statistically significant ($p < 10^{-5}$, *t*-test). Thus, the similarity scores differed between the normal and abnormal groups for SBERT but not for TF-IDF.

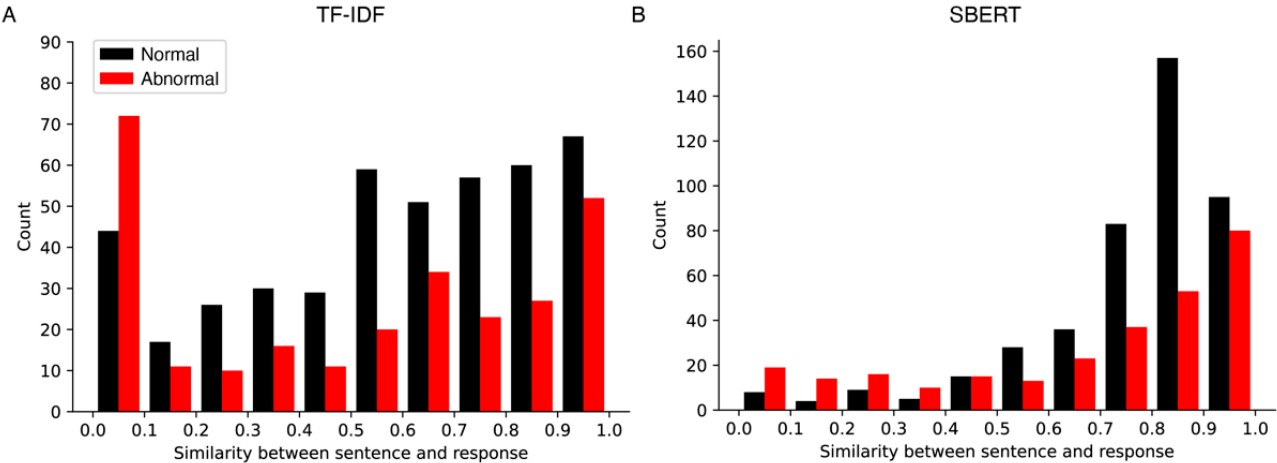

**Figure 2.** Histograms of similarity scores between sentences and responses based on TF-IDF (**A**) and SBERT (**B**).

The similarity scores of the normal participants were higher than those of the abnormal participants, and this difference was significant for some, but not all, of the sentences. The left panel of Figure 3 shows the similarity scores of the normal (black circles) and abnormal (red squares) groups as a function of the sentence ID in the stimulus text, with the error bars representing the standard errors of the mean and the asterisk representing the statistical difference between the normal and abnormal groups. The corresponding *p*-values are shown in the right panel of Figure 3, with the blue horizontal line representing a significance level of 0.05. For both embedding models, the similarity scores of the normal group tended to be higher than those of the abnormal group. However, this difference was more pronounced for SBERT than for TF-IDF. Specifically, the similarity scores based on TF-IDF differ significantly for only one sentence (Figure 3A right). In contrast, the similarity scores based on SBERT differ significantly for five sentences (Figure 3B right). Notably, the sentence (4) that produced a significant difference between the normal and abnormal groups based on TF-IDF did not produce a significant difference between the normal and abnormal groups based on SBERT.

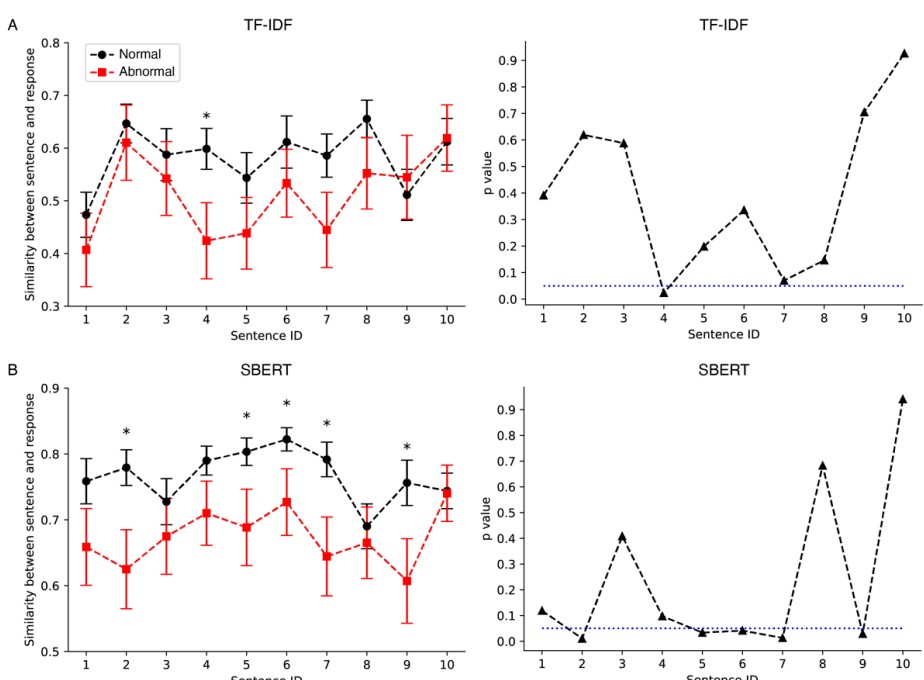

**Figure 3.** The similarity scores (**left**) and *p*-values of comparison between normal and abnormal groups (**right**) based on TF-IDF (**A**) and SBERT (**B**) as function of sentence ID. The asterisk (**left**) represents significant difference between normal and abnormal groups. Blue dashed line (**right**) represents a significance level of 0.05.

### 3.2. Classification Accuracies

The classification accuracies based on SBERT were higher than those based on TF-IDF for all the classifiers, but the accuracy of the different classifiers varied considerably. Figure 4 shows the F1 scores of the seven classifiers for classifying normal and abnormal participants, with the dashed line indicating the F1 score of the baseline classifier (0.56). Error bars represent standard errors of the mean, and the asterisk indicates a significantly higher F1 score compared to that of the baseline (one-tailed *t*-test, $p < 0.05$). Using TF-IDF (light-gray bars in Figure 4), only SVC and random forest classifiers outperformed the baseline classifier significantly. Using the SBERT (dark-gray bars in Figure 4), LDA, SVC, random forest, and KNN classifiers outperformed the baseline classifier significantly. For both embedding models, the random forest classifier produced the highest F1 scores, 0.68 (TF-IDF) and 0.74 (SBERT), with corresponding recall scores of 0.57 (TF-IDF) and 0.61 (SBERT).

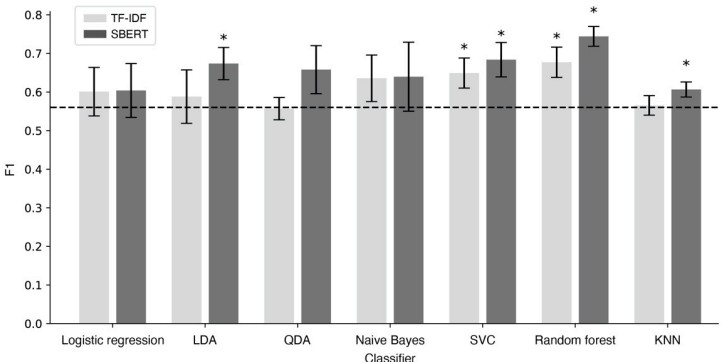

**Figure 4.** F1 scores for classifying normal and abnormal readers using sentence embeddings. The dashed line shows the F1 score of the baseline classifier. Error bars represent standard errors of the mean. The asterisk indicates a significantly higher F1 score compared to that of the baseline (one-tailed *t*-test, $p < 0.05$).

## 4. Discussion

The results of this study show that the similarity score based on the embedding of sentences is an effective way of quantifying reading behavior. The similarity scores between the stimulus text and the responses were lower for poor comprehenders compared to normal students. This is consistent with previous research, indicating that poor comprehenders struggle with understanding the meaning of given sentences and often digress into unrelated topics [3]. It is believed that such behaviors are associated with an attention deficit or inefficient working memory [1,3]. Our method provides quantitative evidence to support this theory.

The main difference between the two embedding models is the use of context in the text and response. TF-IDF is based solely on the frequency of words and is insensitive to their order or relationship to neighboring words. In contrast, the SBERT model is trained to take contextual information into account by considering words around a target word. This difference resulted in qualitatively different sentence embeddings and higher classification accuracies of the SBERT model than those of TF-IDF.

Even with simple linear classifiers, the embedding models produced qualitatively different F1 scores. The F1 scores of the logistic regression classifiers were similar for different embedding models. The F1 score of LDA based on TF-IDF was lower than that of the logistic regression classifier based on TF-IDF. In contrast, the F1 score of LDA based on SBERT was higher than that of the logistic regression classifier based on TF-IDF, resulting in a significantly higher F1 than the baseline classifier. This difference was due to the different assumptions made about the input to the classifiers. LDA assumes that the input is normally distributed. As shown in Figure 2, the distribution of similarity scores based on TF-IDF is bimodal, which does not fit the LDA assumption. In contrast, the distribution of similarity scores based on SBERT is unimodal, which is well modeled by LDA.

Similarly, a comparison of LDA and QDA provides the following insights. Both LDA and QDA assume normality of the input distribution. The only difference is that LDA assumes that the same covariance matrix is shared between the classes, whereas QDA allows each class to have its own covariance matrix, allowing more flexibility in capturing the different distributions of the classes with a larger number of parameters. For both embedding models, the F1 values of the QDA are lower than those of the LDA. Thus, the lower accuracy of the more flexible classifier indicates overfitting.

The F1 values of the Naïve Bayes classifiers were slightly higher than those of logistic regression, but not significantly higher than those of the baseline classifier. This is because the feature dimension (10), which corresponds to the number of sentences in the stimulus text, is relatively low in this study. Thus, scaling the classifier for higher dimensional features at the cost of ignoring the correlation between dimensions provides little gain.

Compared to the Naïve Bayes classifiers, SVM and random forest classifiers produced higher classification accuracies. This suggests that the reader's response to an input text changes as the participant reads through the sentences in the text. Considering this non-stationary behavior allowed the SVM and random forest classifiers to achieve higher classification accuracies.

For both embedding models, the F1 scores of the SVM classifiers were significantly higher than those of the baseline classifier. This is consistent with the previous finding that SVM works well, especially when the number of samples is not enough. The balance between the classification error and sensitivity to outliers can be controlled by the hyper parameter C. With the optimal choice of hyperparameter, SVMs achieved high F1 scores of 0.65 (TF-IDF) and 0.68 (SBERT), with corresponding recall scores of 0.5 (TF-IDF) and 0.46 (SBERT).

The highest accuracy of the random forest classifier demonstrates the effectiveness of the ensemble approach. The random forest classifier is a collection of simple decision trees. Aggregating predictions from multiple trees has improved overall accuracy and reduced overfitting. Consequently, the highest F1 score of 0.74 with a recall score of 0.61 was achieved using SBERT.

Notably, the KNN classifier is the most flexible classifier, but its accuracy is low. This can be attributed to the overfitting phenomenon. The KNN classifier fits complex decision boundaries in the training data and does not generalize well to unseen examples. The drop in classification accuracy was more pronounced for TF-IDF, where the F1 score (0.57) is close to that of the baseline classifier (0.56). In comparison, SBERT is less sensitive to overfitting, as the F1 score of the KNN classifier based on SBERT (0.61) is significantly higher than the baseline score.

Further improvement could be achieved by end-to-end training of a deep neural network. Rather than separating feature extraction and classification, training a deep neural network from end-to-end could lead to higher accuracy by enabling the model to learn both feature extraction and classification tasks simultaneously. However, end-to-end training would require larger amounts of data and is not applicable to the current dataset. The author is in the process of collecting more data to explore this direction in future work.

## 5. Conclusions

In this study, we propose a method to quantify readers' responses to a given text and automate the think-aloud protocol for diagnosing reading comprehension impairments. The data collection procedure of the think-aloud protocol remains the same. We propose the quantification of readers' responses using features based on sentence embedding and standard classification models. The stimulus text and the think-aloud responses were first encoded into high-dimensional vectors using sentence embedding. The similarities between these encoded representations were then used as features. Notably, the similarity scores were lower for poor comprehenders than for average students. Using these similarity-based features, we successfully classified normal from abnormal readers. The highest F1 score of 0.74 was achieved using SBERT embedding in combination with the random forest classifier.

Our future research will focus on further exploring the complex patterns within readers' utterances. In this study, we used pre-trained embedding models to represent the stimulus text and readers' responses for feature (similarity score) extraction and classification. To advance our methodology, our goal is to fine-tune the embedding models using an integrated framework that combines embedding, feature extraction, and classification into a unified model. This integrated model will be trained end-to-end to provide an embedding optimized for the think-aloud protocol. For this purpose, we are actively collecting additional think-aloud data, including texts from diverse topics and corresponding responses of readers. We expect to achieve a higher accuracy in identifying reading comprehension impairments and to uncover individual differences in reading comprehension.

**Funding:** This work was supported by the Soongsil University Research Fund (New Professor Support Research) of 2022.

**Institutional Review Board Statement:** This study was conducted in accordance with the Declaration of Helsinki and approved by the Institutional Review Board of Chonnam National University (1040198-170920-HR-074-02) on 13 October 2017.

**Informed Consent Statement:** A comprehensive written consent form was provided to all participants and their parents. These forms outlined the objectives of the research, the extent of participation, potential benefits and risks, assurances of confidentiality, and the right of the participants to withdraw from the study at their discretion.

**Data Availability Statement:** The data are not publicly available due to Personal Information Protection Act.

**Conflicts of Interest:** The author declares no conflicts of interest.

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
