# Peer review of "Automated Think-Aloud Protocol for Identifying Students with Reading Comprehension Impairment Using Sentence Embedding"

_applsci, doi:10.3390/app14020858_

Round 1

Reviewer 1 Report

Comments and Suggestions for Authors

The paper is an interesting one to demonstrate identifying students with reading comprehension impairment using sentence embedding. However, I have a few comments for a few improvements:

(1) "the richness of the data collected using the think-aloud protocol presents several challenges". In this study, from what aspects to overcome these challenges.

(2) What role does think-aloud protocol play in constructing "feature extraction methods" and "classification models"?

(3) If the use of deep learning model for classification, whether it will produce better results. Whether to optimize the classification model as the direction of further improvement.

Reviewer 2 Report

Comments and Suggestions for Authors

The paper presents the methodology and results in a clear and organized format. Although the data size is relative small (at least not large), data from real patients or people involved in the survey are very valuable and hard to get. So the experiments and results are interesting and important for us to see.

Here are a few comments where the paper should improve to make it more convincing with better readability :

1. Figure 2 and Figure 3 are quite informative but not very convincing for readers to understand the different between the control group and the other group. As the data size is not large, AB test can be used between the similarity scores in the two groups to check the significant difference. And the p values can be checked for demonstrating whether TFIDF or SBERT produces a more significant difference in the similarity scores for the two groups.

2. In addition to 1, a AB test and p-value score can also be calculated for similarity scores for each sentence ID to see which features look much more important. This will add more clear quantitative information to the figure 3. 

3. It would be good to try one or more the state of art sentence embeddings trained by more latest LLM? for example, gtr-large. They are outperforming SBERT in many cases and may have a better performance.

4. Beyond the F1 score, it is meaningful and important to show the recall scores. As in biostatistics, we care more about recall than precision/F1 as we hope we can at least identify all the true patients and send them to the next round of tests for identifying the diseases instead of missing any of them.

Overall, it's a good paper and with the minor changes above, it will highlight the paper and make it more robust and convincing. 
